# Comparison of the selection of nasotracheal tube diameter based on the patient's sex or size of the nasal airway: A prospective observational study

Hye Jin Kim[1], Yunho Roh[2], Soon Young Yun[3], Wyun Kon Park[1], Ha Yan Kim[2], Min Ho Lee[1], Hyun Joo Kim[1]*

1 Department of Anesthesiology and Pain Medicine, Anesthesia and Pain Research Institute, Yonsei University College of Medicine, Seoul, Republic of Korea, 2 Biostatistics Collaboration Unit, Department of Biomedical Systems Informatics, Yonsei University College of Medicine, Seoul, Republic of Korea, 3 Department of Anesthesiology and Pain Medicine, Anesthesia and Pain Research Institute, Gachon University College of Medicine, Incheon, Republic of Korea

* jjollong@gmail.com

**Data Availability Statement:** All relevant data are within the paper and its Supporting information files.

## Abstract

When selecting the nasotracheal tube diameter for nasotracheal intubation, atraumatic introduction of the tube through the nasal passage and a safe location of the tube's cuff and tip should be ensured simultaneously. To maintain safety margin for the tube's cuff and tip from the vocal cords and carina (2 cm and 3 cm, respectively), the maximum allowable proximal-cuff-to-tip distance was calculated as 5 cm less than the measured vocal cords-to-carina distance. The primary aim of this study was to find a single predictive preoperative factor of the nostril size and maximum allowable proximal-cuff-to-tip distance of nasotracheal tubes. The secondary aim was to compare the difference in the safety margin between the maximum allowable proximal-cuff-to-tip distance based on the patient's airway and the actual proximal-cuff-to-tip distance of the selected tube. We used fiberoptic bronchoscope to measure the distance from the vocal cords to the carina for the calculation of the maximum allowable proximal-cuff-to-tip distance. We analyzed the association of preoperative characteristics such as age, sex, height, and weight with the nostril size and maximum allowable proximal-cuff-to-tip distance. The proportion of patients with appropriate locations of both the cuff and tip was evaluated. Sex and height were significant predictive factors of the nostril size and maximum allowable proximal-cuff-to-tip distance, respectively (p = 0.0001 and p = 0.0048). The difference in the safety margin was significantly decreased when the tube diameter was selected based on the nostril size rather than by sex (p<0.0001). The proportion of patients who had the appropriate cuff/tip location was significantly larger (75.2%) when the tube diameter was selected by sex compared to when it was selected by the nostril size (65%) (p<0.0001). It is more suitable to select the nasotracheal tube diameter based on sex rather than by nostril size to ensure the safe location of the tube's cuff and tip simultaneously.

**Funding:** The authors received no specific funding for this work.

**Competing interests:** The authors have declared that no competing interests exist.

## Introduction

Nasotracheal tubes are commonly used to ensure a large oral surgical field during head and neck surgeries. While choosing the size of the nasotracheal tube to be used, three considerations are made. First, the tube diameter should not only be matched with the tracheal diameter [1] but also with the size of the nares; this is because the tube is inserted through the narrow nasal passage, and not through the mouth such as when using an orotracheal tube [2]. Thus, it is recommended to choose a tube with a smaller diameter, compared to the orotracheal tube, to prevent nasal trauma and bleeding [3]. Second, the insertion depth of the tube's cuff and tip from the vocal cords and carina should be appropriate. The nasotracheal tube has a predetermined bend in its structure to guide the outer part of tube toward the head and away from the mouth. This bent portion is usually adjusted at the patient's nose due to the preformed angle, and this tends to determine the insertion depth of the tube's cuff and tip along the patient's airway. Third, surgeries using nasotracheal tubes often need various positions of head flexion, extension, and rotation. These positional changes can cause unpredictable displacements of the tube resulting in undesired movements of the tube's tip and cuff towards or away from the carina and vocal cords [4]. This can cause complications such as bronchial intubation or extubation that could damage the larynx [5, 6]. Therefore, the size of the nasotracheal tube should be selected such that it does not exceed the maximum diameter of the trachea and nasopharyngeal passage, and falls within the safety margin of the distance between the tube's cuff and tip from the vocal cords and carina.

Notably, only a few studies have investigated the parameters for the selection of the tube diameter in adults [1, 2, 7, 8]. Previous studies that have focused on the selection of the largest possible tube diameter performed atraumatic introduction through the nose and trachea [9, 10], or they simply chose the tube diameter by sex [11, 12] without definite references. Findings from other studies indicate that the required depth of the nasotracheal tube could be predicted by the patient's height [13, 14]. All of these previous studies investigated the length or width of the tube separately. However, the length between the tube's cuff and tip increases in proportion to the width [13]; this means that although the size of the nasotracheal tube is appropriate for its width, the length of the tube can be a problem in patients with the carina closer to the vocal cords, as this indicates a short airway [15]. In our previous study, we demonstrated that, in general, more patients have an adequate location of the nasotracheal tube with regard to its placement when the selected size of the tube is decreased, especially in women [16]. These findings are complex and cannot be interpreted in just one direction. As the tube length decreases, a decrease in the bend-to-tip length may reduce the risk of bronchial intubation, while a decrease in the bend-to-cuff length may elevate the risk of extubation. Therefore, it is necessary to determine a suitable method for selecting the tube diameter based on both the width and length. Furthermore, as neck motions can cause the endotracheal tube's tip to move up and down [17], it is recommended that the proximal cuff of the endotracheal tube be located at least 2 cm [16, 18–20] below the vocal cords, while the endotracheal tube's tip should be positioned $5 \pm 2$ cm above the carina with the neck in the neutral position [21]. Thus, in this study, the safety margin for the distance between the vocal cord and proximal cuff was determined to be more than 2 cm, while that for the distance between the tube's tip and carina was more than 3 cm.

The primary aim of this study was to investigate the predictive preoperative factors associated with both the nostril size and maximum allowable proximal-cuff-to-tip distance, which was defined as 50 mm less than the distance between the vocal cords and carina. Based on this result, we analyzed the proportion of patients who would have an adequate tube position within the safety margin of the patient's airway according to the methods of selection of the tube diameter.

## Methods

### Patients

This prospective observational study was conducted at Severance Hospital, Yonsei University Health System, Seoul, Republic of Korea, according to the tenets of the Declaration of Helsinki. The study protocol was approved by the institutional review board of Severance Hospital, Yonsei University Health System, Seoul, Republic of Korea (IRB no. 1-2017-0048, September 6, 2017) and registered at ClinicalTrials.gov (NCT03282604; data collection started on September 25, 2017, and ended on December 6, 2018). Written informed consent was obtained from all patients participating in this study. Between September 2017 and December 2018, we enrolled patients aged 20–70 years (American Society of Anesthesiologists, class I–II) who were scheduled for nasotracheal intubation. We excluded patients who were pregnant, had difficulty breathing, had impaired cervical motion, had undergone emergent surgery, had a nasal disease, or refused to participate in the study.

### Anesthetic management and measurement

After entering the operating room and lying down in the supine position, the patients were asked to block each nostril in turn and identify the one that allowed more comfortable breathing. Standard monitors for pulse oximetry, 3-lead electrocardiography, and non-invasive blood pressure measurement were attached.

Anesthesia was induced using 1–2 mg kg$^{-1}$ propofol (Fresofol 1% MCT; Fresenius Kabi Austria GmbH, Graz, Austria), 0.5–1.0 μg kg$^{-1}$ remifentanil (Ultian; Hanlim Pharm. Co., Ltd., Seoul, Korea), and 0.6 mg kg$^{-1}$ rocuronium (Rocumeron; Ilsung Pharmaceuticals Co., Ltd., Seoul, Korea). Mask ventilation was performed using oxygen at 5 L min$^{-1}$ and sevoflurane 4.0 vol %. Complete muscle relaxation was assessed at the adductor pollicis muscle by a supramaximal train-of-four stimulus applied to the ulnar nerve using a peripheral nerve stimulator (Innervator 252; Fisher & Paykel Healthcare, Auckland, New Zealand). After ensuring complete muscle relaxation, the mask ventilation was stopped and nasopharyngeal airways (PVC airway; SunMed, Grand Rapids, MI, USA) with internal diameters of 6.0, 6.5, 7.0, 7.5, and 8.0 mm, well lubricated with jelly, were inserted through the patient's preferred nostril in the order of increasing size of the airways. The resistance encountered by the nasopharyngeal passage during insertion of the nasopharyngeal airways was recorded as mild or moderate. Smooth insertion with little or slight friction was graded as mild resistance, and obvious resistance with a potential risk of intranasal abrasion was graded as moderate resistance [16]. If moderate resistance was felt, the attempt to insert the nasopharyngeal airway was stopped, and the airway was removed from the patient's nose [3]. The airway's fit relative to the nasopharyngeal passage was assessed based on this resistance. The size of the airway was considered to be correct when mild resistance was felt, and this size was considered as the nostril size for that patient. The airway was considered to be oversized when moderate resistance was felt.

Mask ventilation was resumed for 1 minute to maintain oxygen levels, and a flexible fiberoptic bronchoscope (Olympus LF-GF; Olympus Optical Co., Tokyo, Japan) was inserted into the nasal cavity (Fig 1).

When the bronchoscope approached the carina, tape was applied to the bronchoscope at the nostril. After the bronchoscope was gradually withdrawn until its tip reached the vocal cords, a second piece of tape was applied to the bronchoscope at the nostril [14]. We then removed the bronchoscope from the patient's airway and measured the length between the two pieces of tape; we defined this length as the distance between the vocal cords and carina. During this measurement, mask ventilation was resumed and nasotracheal intubation was

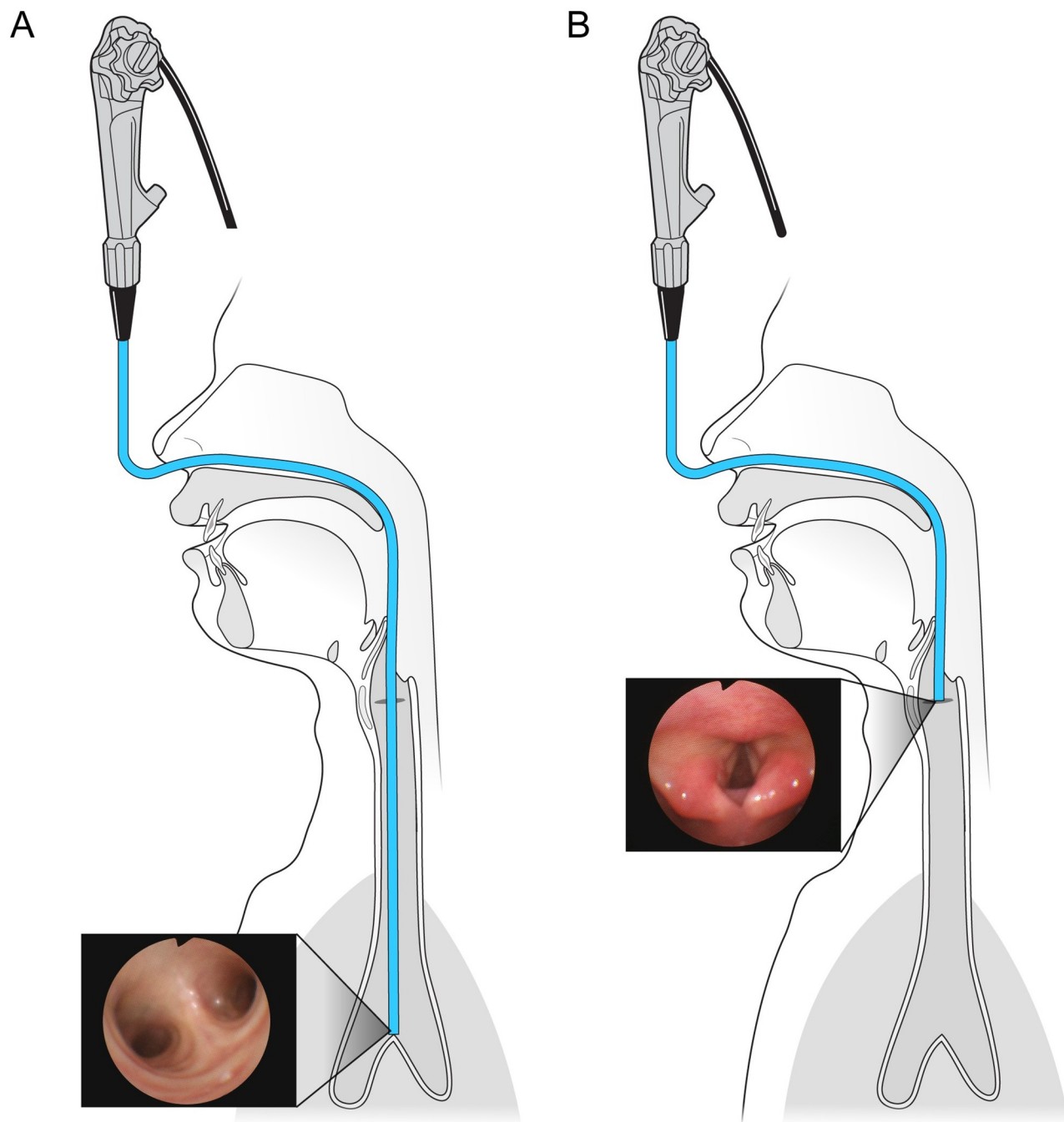

**Fig 1. Measurement of the distance between the carina and vocal cords.** The measurement was performed using a fiberoptic bronchoscope. (A) The fiberoptic bronchoscope was inserted through the nose and towards the carina. (B) The fiberoptic bronchoscope was withdrawn until the vocal cords were just visualized.

performed using an Ivory PVC Portex North Facing Nasal Soft-Seal Cuffed Polar Preformed Endotracheal Tube (Smiths Medical International, Hythe, United Kingdom) and a video laryngoscope (UEScope; UE Medical Devices, Inc. 831 Beacon Street, Suite 136 Newton, MA 02459, USA). The ideal tube position was defined as the position in which the tube's cuff and tip were within the safety margin, with the cuff below the vocal cords and tip above the carina

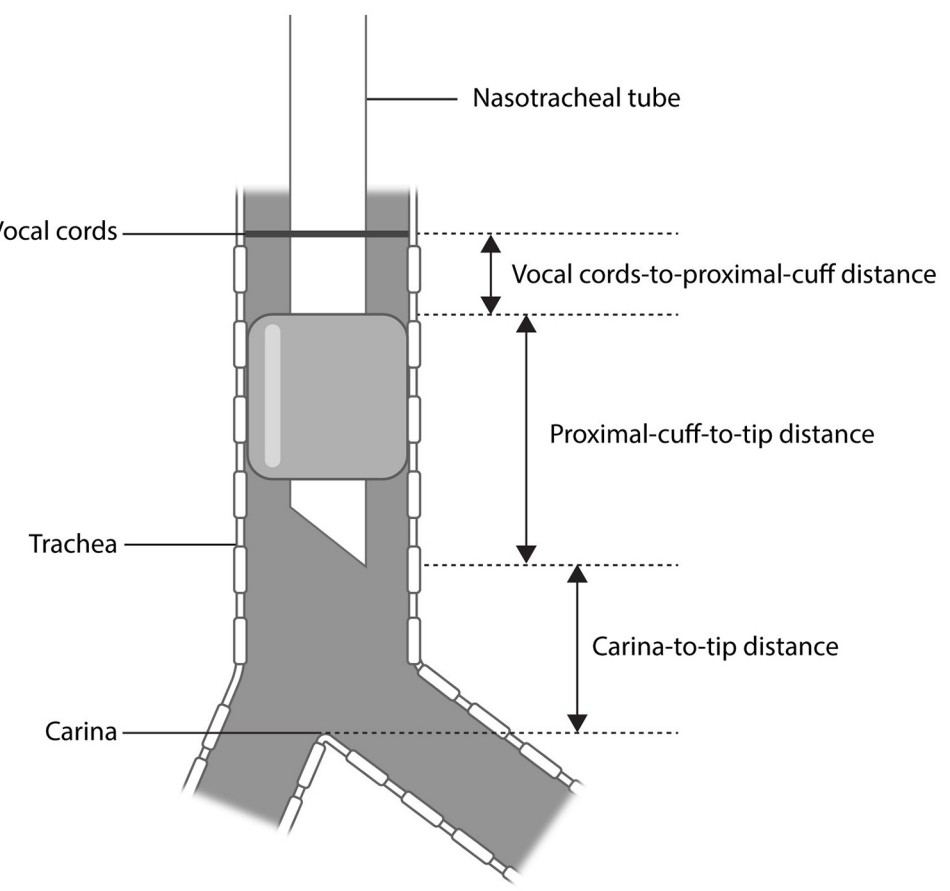

**Fig 2. Insertion depth of the nasotracheal tube's cuff and tip.** The maximum allowable proximal-cuff-to-tip distance was determined as 50 mm less than the calculated distance between the vocal cords and carina; 2 cm between the vocal cords and the tube's proximal cuff and 3 cm between the tube's tip and carina.

despite the various positions that are induced by possible head and neck movements [22, 23] (Fig 2).

To ensure that the selected safety margins for the distances from the tube's proximal cuff and tip to the vocal cords and carina were appropriate, we assumed that the proximal cuff was positioned 2 cm below the vocal cords. Then, we calculated the distance from the tube's tip to the carina and predicted the incidence of achieving an ideal tip position, i.e., more than 3 cm above the carina. A distance between the tip and carina of less than 3 cm despite the proximal cuff being placed 2 cm below the vocal cord indicated that the proximal-cuff-to-tip distance of the tube was too long for the patient. The maximum allowable proximal-cuff-to-tip distance was obtained by subtracting 5 cm, the sum of the safety margins at both extremes, from the distance between the vocal cords and carina.

## Statistical analyses

We investigated the independent variables that could predict the nostril size and maximum allowable proximal-cuff-to-tip distance. A pilot study was carried out using 40 patients to calculate the required sample size. The explanatory power of the regression equation with the variables predicting the appropriate tube length derived from the pilot study was determined as the effect size. Based on the $R^2$ value using G*Power version 3.1 (Faul, Erdfelder, Lang, and

Buchner, Germany), the required sample size was calculated to be 130 (assuming Type I error = 5%, and power = 80%). In total, 145 participants were required after accounting for a 10% dropout rate.

Clinically significant factors among the preoperative characteristics, including age, sex, height, and weight were entered into a multivariable logistic regression model to assess their effect on the maximum allowable proximal-cuff-to-tip distance through a stepwise variable selection. We performed a multivariable ordinal logistic regression analysis for identifying the predictors of nostril size. After identifying the appropriate predictors for nostril size, we aimed to measure the difference in safety margin between the maximum allowable proximal-cuff-to-tip distance and the actual proximal-cuff-to-tip distance according to the two selection methods of tube diameter; based on 1) the measured nostril size, and 2) the identified predictor for nostril size, respectively. The number of patients who had an appropriate distance between the tube's proximal cuff and tip within the maximum allowable distance of the patient's airway was also analyzed and compared for the two selection methods of the tube diameter.

Continuous variables were analyzed using either a two-sided t-test or Mann-Whitney U test based on their normality of distribution, which was determined by the Shapiro-Wilk test. Categorical variables were analyzed using the $\chi^2$ test or Fisher's exact test. Paired data were analyzed using the Wilcoxon signed rank test or McNemar's test. Numerical data are presented as mean ± standard deviation if the data showed normal distribution. If not, the data are presented as median [interquartile range Q1 to Q3]. Categorical variables are presented as the number of patients and percentage. All analyses were performed using the Statistical Package for the Social Sciences (SPSS) version 23 software (IBM Corp., Armonk, NY, USA), R package version 3.4.3 (http://www.R-project.org), and SAS (version 9.4, SAS Inc., Cary, NC, USA).

## Results

Of the 306 patients screened, 146 patients were eventually enrolled. Among them, nine patients dropped out. Overall, 137 patients completed the study (Fig 3) and no missing data were recorded for any patient.

Patient characteristics and airway lengths are shown in Table 1.

In the ordinal logistic regression models to predict the nostril size, sex showed statistically significant correlations (odds ratio, OR = 8.182 [95% confidence interval, CI = 2.825–23.692]; p = 0.0001), while age, height, and weight were not significant. In all women, a tube with a diameter of 6.0 mm was inserted with mild resistance while a tube with a diameter of 6.5 mm was inserted with mild resistance in all men. The maximum allowable proximal-cuff-to-tip distance was 64.1 ± 13.8 mm (69.1 ± 13.5 mm in men and 59.0 ± 12.2 mm in women). In the multivariable logistic regression models to predict the maximum allowable proximal-cuff-to-tip distance, height showed statistically significant correlations (beta coefficient, 0.057; standard error [SE] 0.020; p = 0.0048) while age, sex, and weight did not.

The difference in the safety margin between the maximum allowable proximal-cuff-to-tip distance and actual proximal-cuff-to-tip distance was significantly lesser when the tube diameter was selected by the nostril size than when it was selected by sex (median [Interquartile range]: 4.0 mm [-3.0 mm, 15 mm] and 7.0 mm [0.0 mm, 19 mm], respectively, p<0.0001) (Fig 4).

The number of patients who were expected to have an appropriate maximum allowable proximal-cuff-to-tip distance which exceeded the actual proximal-cuff-to-tip distance was significantly larger when the tube diameter was selected by sex rather than by the nostril size [103 (75.2%) and 89 (65.0%), respectively, p<0.0001] (Fig 5).

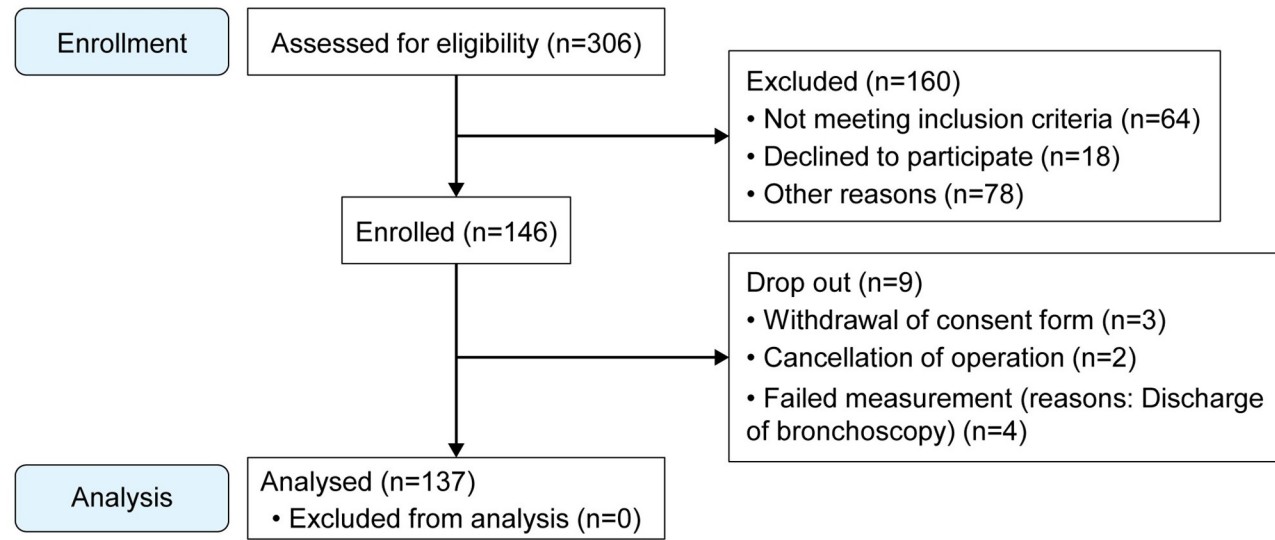

**Fig 3. Flow chart of patient enrolment.** Among the 306 patients scheduled for elective surgery under general anesthesia with nasotracheal intubation between September 2017 and December 2018 at our institution, 64 patients did not meet the inclusion criteria and 96 declined to participate. Among the remaining 146 patients, three withdrew consent, two cancelled the operation, and four had failed measurements. Finally, 137 patients were included.

## Discussion

In this study, we investigated the preoperative predictive factors associated with the selection of both the width and length of the nasotracheal tube. We found that the preoperative patient characteristics which could guide the selection of the size of the nasotracheal tube are not the same for the width and length of the tube, as they are influenced by the patient's sex and height, respectively. This means that it is difficult to find a tube with a diameter that suits the size of the nasal passage and trachea while maintaining an appropriate distance from the vocal cords and carina. Since the company produces tubes of only one length for each value of width, there is a high possibility that the length will not match the tracheal size even if the tube diameter is selected according to the width. Therefore, after choosing a suitable tube width, a better process of guidance must be employed to match the length accordingly in a tube of a given size.

It is generally believed that choosing an appropriate tube diameter and not forcing the tube is essential for preventing complications during nasotracheal intubation [3, 24]. In our study,

**Table 1. Patient characteristics and airway lengths.**

|  | **Men (*n* = 69)** | **Women (*n* = 68)** |
|---|---|---|
| Age (years) | 36 (20–69) | 36 (20–69) |
| Height (cm) | 173.0 ± 5.8 | 160.5 ± 5.0 |
| Weight (kg) | 71.2 ± 9.3 | 57.8 ± 11.0 |
| Body mass index (kg m$^{-2}$) | 23.8 ± 2.4 | 22.4 ± 3.6 |
| Preferred nostril side, left/right/both | 18/41/10 | 16/34/18 |
| Nostril size (mm) | 7.0 [7.0 to 7.5] | 6.5 [6.0 to 7.0] |
| Length from the vocal cords to the carina (cm) | 11.9 ± 1.4 | 10.9 ± 1.2 |

Data are presented as mean (range) for age, number of patients for preferred nostril size, mean ± standard deviation, or median [interquartile range Q1 to Q3].

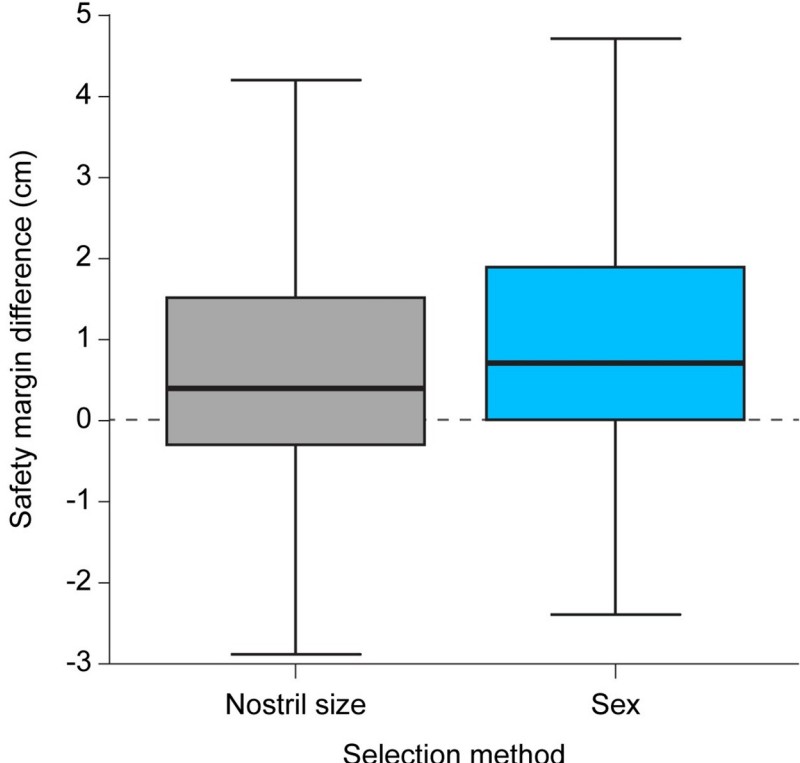

**Fig 4. Difference in the safety margin between the maximum allowable proximal-cuff-to-tip distance and actual proximal-cuff-to-tip distance.** The selection of the tube diameter was based on nostril size or sex. A negative value indicated that the length from the tube's proximal cuff to its tip was too long. This means that the actual proximal-cuff-to-tip distance of the selected tube exceeded the maximum allowable proximal-cuff-to-tip distance; the tip was too close to the carina for maintaining 2 cm safety margin for the distance between the vocal cords and the tube's proximal cuff.

the size of the nasopharyngeal airway with slight resistance during insertion was defined as the nostril size. This is based on previous studies which have demonstrated that nasal passages offering moderate resistance and a longer nasal passage time were associated with more frequent epistaxis rather than those with slight resistance [12, 25]. Although preanesthetic evaluations such as fiberoptic nasoendoscopy can be accurate for evaluating the presence of nasal abnormalities, in clinical practice, routine fiberoptic evaluations are expensive and cumbersome [26].

In our study, the preoperative predictive factor associated with the nostril size was sex, which has been shown to influence the structure of the anatomic airway (e.g., tracheal diameter) [1]. Women are at a greater risk of facing postextubation laryngeal edema as their larynx is smaller and their mucosal membrane is more vulnerable than that in men [27]. In our study, tubes with internal diameters of 6.0 mm and 6.5 mm fit well within the nasopharyngeal passages of all women and men, respectively. This size is smaller than that recommended while selecting an orotracheal tube which is a tube diameter of 6.5–7.0 mm for women and 7.0–7.5 mm for men [1]. However, the nasotracheal tube needs to be the same size or two sizes smaller than the corresponding orotracheal tube based on recommendations [3], and a tube diameter of 6.0–6.5 mm matches the tracheal diameter of adults fairly well [1, 2, 22]. Moreover, a previous study showed that the use of an endotracheal tube with a diameter exceeding 7.5 mm was associated with a 27.6 times higher rate of tracheal stenosis compared to that with a tube

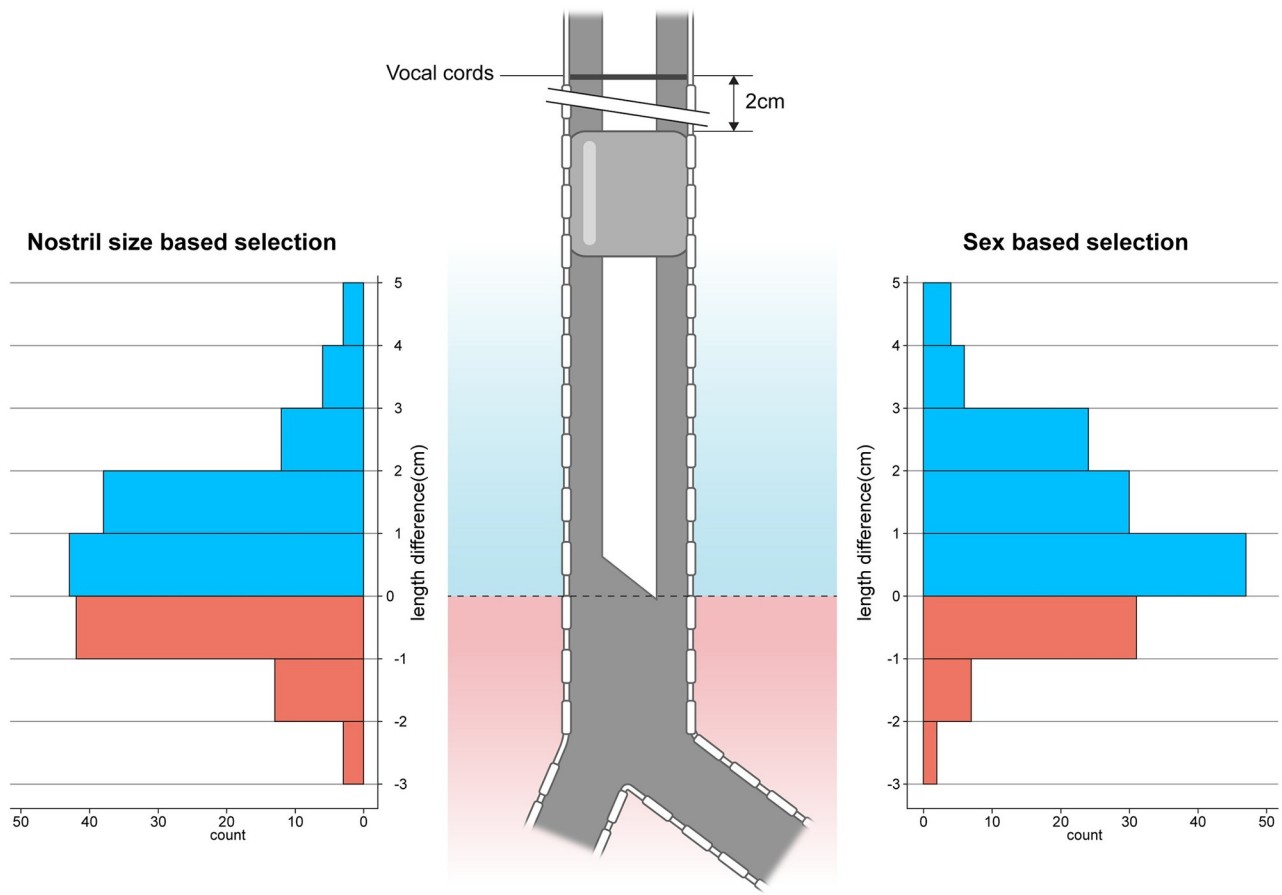

**Fig 5. Nostril-size-based vs. sex-based selection.** The number of patients expected to have the appropriate tip location when the tube sizes were selected according to the nostril size or sex, with the tube's proximal cuff located 2 cm below the vocal cords. A red area means that the tube's tip was too close to the carina causing a risk of endobronchial intubation in various neck positions.

diameter equal to or less than 7.5 mm [28]. Additionally, at a tube width of 6.0 mm, it has been shown that the airway pressure and resistance is suitable for providing adequate ventilation [29], and a smaller-sized tube is better for decreasing air leaks around the cuff by decreasing redundant folds of the cuff [30] and reducing the incidence of postoperative sore throat [31]. Therefore, the approach of selecting the nasotracheal tube diameter based on sex can be recommended since sex is a significant predictive factor for nostril size, and a tube diameter of 6.0–6.5 mm could potentially be an excellent choice for reducing complications due to an oversized tube.

The problem of depth is secondary to that of width as the tube is first inserted into the patient's nasopharyngeal passage. After choosing the appropriate tube diameter, the nasotracheal tube should be inserted into the appropriate position as head and neck surgery involves various neck positions and it is difficult to approach and manage the tube during the surgery. The changes in neck position can cause changes in the tracheal length [32], and the direction of tube displacement is usually cephalad with head extension and caudal with head flexion, although head extension caused unintended right main stem intubation in 4% of patients in a previous study [4]. Neck flexion and extension resulted in a tube displacement of more than 10 mm in more than 70% of patients in another study [17]. Head rotation displaces the tube in various directions such as up or down [4], and tube displacements up to 17 mm occurred in

33% of patients in the abovementioned study [17]. Therefore, placing the tube securely into the safe area is mandatory considering the degree and direction of tube movements are often unpredictable [4]. In this study, we set the required minimum safety margin at 20 mm and 30 mm for the vocal cords-to-proximal-cuff distance and carina-to-tip distance, respectively, based on previous studies [21, 22].

Previous studies evaluating nasotracheal tubes have investigated the distance between the nares and vocal cords [13, 14]. The distance from the starting point of the nares is considered based on the assumption that the predetermined bend will be placed on the nares. The advance or withdrawal of the nasotracheal tube beyond this predetermined bend was possible because the material of the tube which was used in this study was soft and flexible [2, 33]. This meant that the predetermined bend could move backwards or forwards beyond the patient's nose, and previous studies that examined only the length of the tube separately assumed that the position of these tubes could be adjusted as needed [13, 14]. In this study, it was difficult to fix the proximal cuff 2 cm below the vocal cords in four men because of the short bend-to-proximal-cuff length when the tube diameter was selected by sex. However, it is only necessary to advance the black mark on the predetermined bend into the nostrils less than 1 cm. Therefore, the distance between the vocal cords and carina is clinically more critical than the distance from the nares to the vocal cords or carina [15]. In our study, we investigated the predictive factors associated with the selection of the optimal proximal-cuff-to-tip distance, based on the distance between the vocal cords and carina as measured in the patient's airway. We confirmed that height is the significantly associated factor here, similar to findings from previous studies [13, 15] although these studies only evaluated the tube length and not the tube's internal diameter.

We clearly showed that the tube depth and thickness cannot be determined together, and that they must be determined separately. Therefore, we analyzed whether the tube's tip could be placed and maintained within the safety margin under the various possible neck positions while the tube's proximal cuff was correctly positioned below the vocal cords, based on our measurement of the tracheal length. We compared the two methods of selection of the tube thickness based on sex and the nostril size. We found that even after the selection of the appropriate tube thickness, the depths of the inserted tube's proximal cuff and tip can fall outside the safety margin in about 30% of all patients. We found that the sex-based selection method was significantly better with respect to the safety margin of the distance between the tube's proximal cuff and tip, compared with the selection method based on the measured nostril size. This is because the shorter tube has a shorter proximal-cuff-to-tip distance, which allows a safer distance to be maintained between the vocal cords and carina compared to a longer tube [13]. Therefore, the production of tubes with various internal diameters and lengths in the future is necessary, but until then, it would be advisable to select the diameter of the nasotracheal tube diameter by sex within the range of 6.0–6.5 mm.

There are several limitations in this study. First, we defined the optimal proximal cuff and tip location to be 2 cm below the vocal cords and 3 cm above the carina. If the safety margin was applied differently [34], the proportion of patients would have been different. However, we believe that even with a different safety margin, the selection method based on sex would still be more appropriate compared to that based on the measured nostril size. Second, we did not measure the changes in the airway's length that occurred along with changes in head position. Rather, we tried to find a way to evaluate whether this commercially available tube could be located within a safe area along the patient's airway, and to select a tube diameter that would increase the safety margin. Third, nasotracheal tubes can have various sizes and lengths, and the number of patients who have airways determined to be of a safe length for the given tube diameter needs to be evaluated for other types of tubes. The bend-to-cuff and proximal

cuff-to-tip lengths are different among commercially available tubes [13]. Fourth, we did not investigate the adequacy of the bend-to-cuff length. The fact that the tube used in this study had a relatively longer bend-to-cuff length than that of other commercially available tubes [13], coupled with the softness of the bent portion, allowed us to position the proximal cuff 2 cm below the vocal cords in this study population. However, increasing the bend-to-cuff length by 1 cm could eliminate concerns about advancing the tube beyond predetermined bend in our study.

In conclusion, when selecting the nasotracheal tube diameter based on sex, the proximal cuff and tip of the tube are more likely to stay within the safety margin for the distance from the vocal cords and carina than during the selection of the tube diameter based on the largest internal diameter of nostril size, thus precluding the potential occurrence of bronchial intubation or accidental extubation. To maximize safety, it is necessary to choose the tube diameter based on sex and avoid the fitting of oversized tubes into the nasal passage, as well as to shorten the length between the tube's proximal cuff and tip.

## Supporting information

**S1 Checklist. STROBE checklist.**
(DOCX)

**S1 Data. Dataset of this study.**
(XLSX)

**S1 Text. Study details.**
(DOCX)

## Acknowledgments

The authors would like to thank Dong-Su Jang, MFA (Medical Illustrator, Medical Research Support Section, Yonsei University College of Medicine, Seoul, South Korea), for his help with the illustrations.

## Author Contributions

**Conceptualization:** Hye Jin Kim, Yunho Roh, Soon Young Yun, Wyun Kon Park, Ha Yan Kim, Min Ho Lee, Hyun Joo Kim.

**Data curation:** Hye Jin Kim, Yunho Roh, Soon Young Yun, Wyun Kon Park, Ha Yan Kim, Min Ho Lee, Hyun Joo Kim.

**Formal analysis:** Hye Jin Kim, Yunho Roh, Soon Young Yun, Wyun Kon Park, Ha Yan Kim, Min Ho Lee, Hyun Joo Kim.

**Investigation:** Hye Jin Kim, Yunho Roh, Soon Young Yun, Wyun Kon Park, Min Ho Lee, Hyun Joo Kim.

**Methodology:** Hye Jin Kim, Yunho Roh, Soon Young Yun, Wyun Kon Park, Ha Yan Kim, Min Ho Lee, Hyun Joo Kim.

**Project administration:** Hye Jin Kim, Soon Young Yun, Wyun Kon Park, Hyun Joo Kim.

**Resources:** Hye Jin Kim, Wyun Kon Park, Ha Yan Kim, Hyun Joo Kim.

**Software:** Hye Jin Kim, Yunho Roh, Ha Yan Kim, Hyun Joo Kim.

**Supervision:** Yunho Roh, Hyun Joo Kim.

**Validation:** Hye Jin Kim, Yunho Roh, Ha Yan Kim, Hyun Joo Kim.

**Visualization:** Hye Jin Kim, Yunho Roh, Hyun Joo Kim.

**Writing – original draft:** Hye Jin Kim, Hyun Joo Kim.

**Writing – review & editing:** Hye Jin Kim, Yunho Roh, Soon Young Yun, Wyun Kon Park, Ha Yan Kim, Min Ho Lee, Hyun Joo Kim.

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
