## [Decision Letter · Decision Letter 0]

20 Jan 2021

PONE-D-20-38950

Comparison of selection of nasotracheal tube diameter based on patient’s sex or size of the airway: A Prospective Observational Study

PLOS ONE

Dear Dr. HYUN JOO KIM

Thank you for submitting your manuscript to PLOS ONE. After careful consideration, we feel that it has merit but does not fully meet PLOS ONE’s publication criteria as it currently stands. Therefore, we invite you to submit a revised version of the manuscript that addresses the points raised during the review process.

We look forward to receiving your revised manuscript.

Kind regards,

Ehab Farag, MD FRCA FASA

Academic Editor

PLOS ONE

Journal Requirements:

2.Thank you for stating the following financial disclosure:

Additional Editor Comments:

I would appreciate if you pay careful attention in your response to the reviewers' comments. 

Reviewers' comments:

Reviewer's Responses to Questions

**Comments to the Author**

1. Is the manuscript technically sound, and do the data support the conclusions?

Reviewer #1: Yes

Reviewer #2: Yes

2. Has the statistical analysis been performed appropriately and rigorously? 

Reviewer #1: I Don't Know

Reviewer #2: I Don't Know

3. Have the authors made all data underlying the findings in their manuscript fully available?

Reviewer #1: Yes

Reviewer #2: Yes

4. Is the manuscript presented in an intelligible fashion and written in standard English?

Reviewer #1: Yes

Reviewer #2: No

5. Review Comments to the Author

Reviewer #1: As a physician that performs frequent nasal intubation I was excited to review this paper and examine the outcomes found. However I found the reading of the paper cumbersome and difficult to follow. I found the distances measured (tube and trachea) to be ill defined and confusion.

I feel if the "margin of safety" which was mentioned multiple times was defined more clearly at the beginning of the paper this would have helped. The same goes for " maximum allowable distance - please define clearly early in the paper.

It may be helpful to use a formula to determine maximum allowable distance for example is maximum allowable distance the distance between the cords and carina minus 50 mm?

some specific notes

The conclusion that both sex and norstril size needs to be taken into consideration when selecting tube size is interesting. what therefore is the best method for selecting tube size?

No mention was made on tubes that end up being too short and end up with the cuff above the vocal cord.

line 29-30 - a fiberoptic scope was used to measure the distance from the vocal cords to the carina and to calculate the max allowable distance

should this read the measurement found was used to calculate the max allowable distance.

line 81-82 - the tip is likely to fall outside the safety margin despite the cuff moving away from the vocal cords increasing the level of safety

could this be rephrased as it does not make sense to me

Was any sort of vasocostrictive agent used to minimize nasal bleeding during intubation

When the patients were intubated was this performed with fob or direct laryngoscopy?

line 147 - various positions that are induced by possible head and neck movement - how did you determine movement of the tube with head position changes

- line 153-155 - how did you perform these measurements

- use consistnet measurements - in one sentence you used mm (line 154) and another instance you used cm (234)

line 232-234 - sentence needs re phrasing

line 308-310 - did you actually measure tube movement with different head and neck movements? or speculate that the tube moved

Table 1 - nostril size - units?

Reviewer #2: You address a real problem The current design and sizing on nasal endotracheal tubes leaves a lot to be improved on. Commercially available tubes are frequently either too wide in diameter or short in length for our patients.

This paper is not easy to read. Choice of terms leads to confusion

The title implies that you will be comparing selection of ETT based on either sex of patient or size of nasal airway what the patient will accommodate. (Actually the term "nasal" is not in title and should be included)

Many people use nasal decongestants as a premedication for nasal intubations. They will cause the nasal mucous membranes to shrink and enlarge the size of the nasal airway to accommodate a larger tube. Please comment.

Your use of the term cuff-to-tip is confusing. Is this proximal cuff to tip or distal cuff to tip?

Maximum "allowable" tube diameter is unclear and confusing. Why not say - "Appropriate tube sizing based on measurement"

The term "adhesive plasters" is not commonly used. Perhaps "tape" would be better?

On Line 146 Ideal position is when cuff is below the vocal cords "and" tip of tube is above the carina . Not - "or"

There needs to be a clearer discussion of what you are measuring and the clinical implications of each measurement. From the best that I can understand, you are saying that if you place tube based on sex of patient you place 6.0 in female and 6.5 in male. If you place based on sizing then you place 6.5 in female and 7.0 in male. From the text it is not clear to me what the distance from the cords and carina is each scenario. Please reword and make in clearer

Questions not discussed in paper that I would like answered

1. Are all commercially available naso-tracheal tubes identical? Is each size off ETT the exact same length? Is the bend in the tube is the exact same place? Is the distance from the proximal end of the cuff to the tip of the tube the same in all tubes? RAE tubes? Parker Tubes?

6. PLOS authors have the option to publish the peer review history of their article (what does this mean?). If published, this will include your full peer review and any attached files.

Reviewer #1: No

Reviewer #2: No

---

## [Author Response · Author response to Decision Letter 0]

20 Feb 2021

Reviewer #1: As a physician that performs frequent nasal intubation I was excited to review this paper and examine the outcomes found. However I found the reading of the paper cumbersome and difficult to follow. I found the distances measured (tube and trachea) to be ill defined and confusion.

1) I feel if the "margin of safety" which was mentioned multiple times was defined more clearly at the beginning of the paper this would have helped. The same goes for " maximum allowable distance - please define clearly early in the paper.

Response: We thank the reviewer for their valuable suggestion, and apologize for the confusion. The maximum allowable proximal cuff-to-tip distance is explained in Fig. 2, and the suggested definitions have been added to the introduction section of the revised manuscript as follows: (Lines 87-89)

“The primary aim of this study was to investigate the predictive preoperative factors associated with both the nostril size and maximum allowable proximal-cuff-to-tip distance, which was defined as 50 mm less than the distance between the vocal cords and carina.”

(Lines 80-86)

“Furthermore, as neck motions can cause the endotracheal tube’s tip to move up and down [17], it is recommended that the proximal cuff of the endotracheal tube be located at least 2 cm [16, 18-20] below the vocal cords, while the endotracheal tube’s tip should be positioned 5 ± 2 cm above the carina with the neck in the neutral position [21]. Thus, in this study, the safety margin for the distance between the vocal cord and proximal cuff was determined to be more than 2 cm, while that for the distance between the tube’s tip and carina was more than 3 cm.”

2) It may be helpful to use a formula to determine maximum allowable distance for example is maximum allowable distance the distance between the cords and carina minus 50 mm?

some specific notes

Response: We thank the reviewer for their pertinent suggestion. Based on this advice, we have used the formula suggested by you to determine the maximum allowable proximal-cuff-to-tip distance, and we have added this to the introduction section as follows: (Lines 87-89)

“The primary aim of this study was to investigate the predictive preoperative factors associated with both the nostril size and maximum allowable proximal-cuff-to-tip distance, which was defined as 50 mm less than the distance between the vocal cords and carina.”

3) -The conclusion that both sex and norstril size needs to be taken into consideration when selecting tube size is interesting. what therefore is the best method for selecting tube size?

Response: We thank the reviewer for their comment. We believe that the best method for selecting tube diameter is based on sex, which is a predictive factor of nostril size, rather than based on the measured nostril size of each individual. 

To reach this conclusion, we first confirmed that sex was a predictive factor of nostril size. After that, we analyzed the number of patients expected to have the appropriate tip location when the tube sizes were selected according to the nostril size or sex, with the tube’s proximal cuff located 2 cm below the vocal cords. As a result, we confirmed that selecting the tube diameter according to sex increases the possibility of appropriate tube positioning. 

We tried to avoid any misunderstanding by unifying these terms throughout the revised manuscript. We apologize for the confusion.

4) No mention was made on tubes that end up being too short and end up with the cuff above the vocal cord

Response: We thank the reviewer for their pertinent comment. Based on their suggestion, we have modified the discussion section as follows: (Lines 309-320)

“Previous studies evaluating nasotracheal tubes have investigated the distance between the nares and vocal cords [13, 14]. The distance from the starting point of the nares is considered based on the assumption that the predetermined bend will be placed on the nares. The advance or withdrawal of the nasotracheal tube beyond this predetermined bend was possible because the material of the tube which was used in this study was soft and flexible [2, 33]. This meant that the predetermined bend could move backwards or forwards beyond the patient’s nose, and previous studies that examined only the length of the tube separately assumed that the position of these tubes could be adjusted as needed [13, 14]. In this study, it was difficult to fix the proximal cuff 2 cm below the vocal cords in four men because of the short bend-to-proximal-cuff length when the tube diameter was selected by sex. However, it is only necessary to advance the black mark on the predetermined bend into the nostrils less than 1 cm.”

(Lines 353-359)

“Fourth, we did not investigate the adequacy of the bend-to-cuff length. The fact that the tube used in this study had a relatively longer bend-to-cuff length than that of other commercially available tubes [13], coupled with the softness of the bent portion, allowed us to position the proximal cuff 2 cm below the vocal cords in this study population. However, increasing the bend-to-cuff length by 1 cm could eliminate concerns about advancing the tube beyond predetermined bend in our study.”

5) line 29-30 - a fiberoptic scope was used to measure the distance from the vocal cords to the carina and to calculate the max allowable distance

should this read the measurement found was used to calculate the max allowable distance.

Response: We thank the reviewer for their important comment. The reviewer has understood correctly, and we apologize for the confusion caused due to the unclear wording in our manuscript. Therefore, we have revised this portion as follows to improve the clarity: (Lines 31-33)

“We used fiberoptic bronchoscope to measure the distance from the vocal cords to the carina for the calculation of the maximum allowable proximal-cuff-to-tip distance”

6) line 81-82 - the tip is likely to fall outside the safety margin despite the cuff moving away from the vocal cords increasing the level of safety

could this be rephrased as it does not make sense to me

Response: We thank the reviewer for their comment. We apologize for the confusion. This sentence has been modified as follows: (Lines 76-78)

“These findings are complex and cannot be interpreted in just one direction. As the tube length decreases, a decrease in the bend-to-tip length may reduce the risk of bronchial intubation, while a decrease in the bend-to-cuff length may elevate the risk of extubation.”

7) Was any sort of vasoconstrictive agent used to minimize nasal bleeding during intubation

Response: We thank the reviewer for their question. Unfortunately, in the Republic of Korea, it is difficult to prescribe a nasal spray to each individual to minimize bleeding due to problems related to health insurance. For patients with coagulopathy, such a spray is selectively used at the expense of the hospital. In conclusion, vasoconstrictive agents were not used in our study.

When the patients were intubated was this performed with fob or direct laryngoscopy?

Response: We thank the reviewer for their question. We used a videolaryngoscope for intubation. We have revised the methods section to reflect this as follows: (Lines 150-155)

“During this measurement, mask ventilation was resumed and nasotracheal intubation was performed using an Ivory PVC Portex North Facing Nasal Soft-Seal Cuffed Polar Preformed Endotracheal Tube (Smiths Medical International, Hythe, United Kingdom) and a videolaryngoscope (UEScope; UE Medical Devices, Inc. 831 Beacon Street, Suite 136 Newton, MA 02459, USA).”

8)line 147 - various positions that are induced by possible head and neck movement - how did you determine movement of the tube with head position changes

Response: We thank the reviewer for their question. We determined this movement based on our reference to previous papers. We have added the following information to the introduction section of our revised manuscript: 

(Lines 80-86)

“Furthermore, as neck motions can cause the endotracheal tube’s tip to move up and down [17], it is recommended that the proximal cuff of the endotracheal tube be located at least 2 cm [16, 18-20] below the vocal cords, while the endotracheal tube’s tip should be positioned 5 ± 2 cm above the carina with the neck in the neutral position [21]. Thus, in this study, the safety margin for the distance between the vocal cord and proximal cuff was determined to be more than 2 cm, while that for the distance between the tube’s tip and carina was more than 3 cm.”

9) line 153-155 - how did you perform these measurements

Response: We thank the reviewer for their question. We have clarified the manner in which we performed these measurements by revising the methods section as follows: (Lines 160-168)

“To ensure that the selected safety margins for the distances from the tube’s proximal cuff and tip to the vocal cords and carina were appropriate, we assumed that the proximal cuff was positioned 2 cm below the vocal cords. Then, we calculated the distance from the tube’s tip to the carina and predicted the incidence of achieving an ideal tip position, i.e., more than 3 cm above the carina. A distance between the tip and carina of less than 3 cm despite the proximal cuff being placed 2 cm below the vocal cord indicated that the proximal-cuff-to-tip distance of the tube was too long for the patient. The maximum allowable proximal-cuff-to-tip distance was obtained by subtracting 5 cm, the sum of the safety margins at both extremes, from the distance between the vocal cords and carina.

10) use consistnet measurements - in one sentence you used mm (line 154) and another instance you used cm (234)

Response: We thank the reviewer for their valuable suggestion and apologize for this inconsistency. We have used ‘cm’ in both the instances in the revised manuscript.

11)line 232-234 - sentence needs re phrasing

Response: We thank the reviewer for their valuable comment. We have modified this sentence as follows: (Lines 247-251)

“The number of patients expected to have the appropriate tip location when the tube sizes were selected according to the nostril size or sex, with the tube’s proximal cuff located 2 cm below the vocal cords.”

12) line 308-310 - did you actually measure tube movement with different head and neck movements? or speculate that the tube moved

Response: We thank the reviewer for their comment. We speculated that this movement occurred. We have modified the discussion as follows: (Lines 329-332)

“Therefore, we analyzed whether the tube’s tip could be placed and maintained within the safety margin under the various possible neck positions while the tube’s proximal cuff was correctly positioned below the vocal cords, based on our measurement of the tracheal length.”

13)Table 1 - nostril size - units? 

Response: We thank the reviewer for their comment. We have added the units (mm) to the table. We are very grateful for the reviewer’s constructive advice. Based on these suggestions, we believe that the content is much clearer in the revised manuscript.

Reviewer #2: You address a real problem The current design and sizing on nasal endotracheal tubes leaves a lot to be improved on. Commercially available tubes are frequently either too wide in diameter or short in length for our patients.

This paper is not easy to read. Choice of terms leads to confusion

1) The title implies that you will be comparing selection of ETT based on either sex of patient or size of nasal airway what the patient will accommodate. (Actually the term "nasal" is not in title and should be included)

Response: We thank the reviewer for their valuable comments. We have modified the title as follows:

“Comparison of the selection of nasotracheal tube diameter based on the patient’s sex or size of the nasal airway: a prospective observational study”

2) Many people use nasal decongestants as a premedication for nasal intubations. They will cause the nasal mucous membranes to shrink and enlarge the size of the nasal airway to accommodate a larger tube. Please comment.

Response: We thank the reviewer for their question. Unfortunately, in the Republic of Korea, it is difficult to prescribe a nasal spray to each individual due to problems related to health insurance. For patients with coagulopathy, such a spray is selectively used at the expense of the hospital. In conclusion, vasoconstrictive agents were not used in our study.

3) Your use of the term cuff-to-tip is confusing. Is this proximal cuff to tip or distal cuff to tip?

Response: We thank the reviewer for their comment. We have revised the term as ‘proximal-cuff-to-tip’ throughout the manuscript to avoid confusion.

4) Maximum "allowable" tube diameter is unclear and confusing. Why not say - "Appropriate tube sizing based on measurement"

Response: We thank the reviewer for their comment. The maximum "allowable" tube diameter was modified to ‘nostril size’ or ‘measured nostril size’ depending on the context in order to maintain consistency with figures 4 and 5.

5) The term "adhesive plasters" is not commonly used. Perhaps "tape" would be better?

Response: We thank the reviewer for their valuable suggestion. We have modified the methods section based on this advice as follows: (Lines 142-144)

“When the bronchoscope approached the carina, tape was applied to the bronchoscope at the nostril. After the bronchoscope was gradually withdrawn until its tip reached the vocal cords, a second piece of tape was applied to the bronchoscope at the nostril [14].”

6) On Line 146 Ideal position is when cuff is below the vocal cords "and" tip of tube is above the carina . Not - "or"

Response: We thank the reviewer for their valuable suggestion. We have modified this line as follows: (Lines 150-153)

“The ideal tube position was defined as the position in which the tube’s cuff and tip were within the safety margin, with the cuff below the vocal cords and tip above the carina despite the various positions that are induced by possible head and neck movements”

7) There needs to be a clearer discussion of what you are measuring and the clinical implications of each measurement. From the best that I can understand, you are saying that if you place tube based on sex of patient you place 6.0 in female and 6.5 in male. If you place based on sizing then you place 6.5 in female and 7.0 in male. From the text it is not clear to me what the distance from the cords and carina is each scenario. Please reword and make in clearer

Response: We thank the reviewer for their pertinent suggestion. We apologize for the confusion. We compared two selection methods of tube diameter, based on 1) the measured nostril size, and 2) the identified predictor for nostril size, i.e., sex. When the selection was based on the nostril size, the tube diameter corresponding to the measured nostril size was selected for each individual. When the selection was based on sex, 6.0 for women and 6.5 for men, which were all allowable in each sex, were used. We have unified the use of these terms throughout the text, and have revised the introduction, methods, and discussion sections to improve the clarity. We have revised the statistical analysis section as follows: (Lines 179-187)

“Clinically significant factors among the preoperative characteristics, including age, sex, height, and weight were entered into a multivariable logistic regression model to assess their effect on the maximum allowable proximal-cuff-to-tip distance through a stepwise variable selection. We performed a multivariable ordinal logistic regression analysis for identifying the predictors of nostril size. After identifying the appropriate predictors for nostril size, we aimed to measure the difference in safety margin between the maximum allowable proximal-cuff-to-tip distance and the actual proximal-cuff-to-tip distance according to the two selection methods of tube diameter; based on 1) the measured nostril size, and 2) the identified predictor for nostril size, respectively.” 

Questions not discussed in paper that I would like answered

1. Are all commercially available naso-tracheal tubes identical? Is each size off ETT the exact same length? Is the bend in the tube is the exact same place? Is the distance from the proximal end of the cuff to the tip of the tube the same in all tubes? RAE tubes? Parker Tubes?

Response: We thank the reviewer for their pertinent questions. The bend-to-cuff and proximal-cuff-to-tip lengths are different among commercially available tubes for each size. Portex tubes have a relatively longer bend-to-cuff length compared to other commercially available tubes with a soft bend portion. In contrast, RAE tubes have a hard bend portion and short bend-to-cuff length. Unfortunately, Parker tubes are not available in the Republic of Korea.

Based on your insightful advice, we believe that we were able to improve our manuscript considerably. We have also revised the limitation section accordingly: (Lines 352-359)

“The bend-to-cuff and proximal cuff-to-tip lengths are different among commercially available tubes [13]. Fourth, we did not investigate the adequacy of the bend-to-cuff length. The fact that the tube used in this study had a relatively longer bend-to-cuff length than that of other commercially available tubes [13], coupled with the softness of the bent portion, allowed us to position the proximal cuff 2 cm below the vocal cords in this study population. However, increasing the bend-to-cuff length by 1 cm could eliminate concerns about advancing the tube beyond predetermined bend in our study.”

---

## [Editor Report · Decision Letter 1]

24 Feb 2021

Comparison of the selection of nasotracheal tube diameter based on the patient’s sex or size of the nasal airway: a prospective observational study

PONE-D-20-38950R1

Dear Dr.

   HYUN JOO KIM 

We’re pleased to inform you that your manuscript has been judged scientifically suitable for publication and will be formally accepted for publication once it meets all outstanding technical requirements.

Kind regards,

Ehab Farag, MD FRCA FASA

Academic Editor

PLOS ONE
---

## [Editor Report · Acceptance letter]

26 Feb 2021

PONE-D-20-38950R1 

Comparison of the selection of nasotracheal tube diameter based on the patient’s sex or size of the nasal airway: a prospective observational study 

Dear Dr. KIM:

I'm pleased to inform you that your manuscript has been deemed suitable for publication in PLOS ONE. Congratulations! Your manuscript is now with our production department. 

Kind regards, 

on behalf of

Dr. Ehab Farag 

Academic Editor

PLOS ONE